# Enhancing Underwater Images via Color Correction and Multiscale Fusion

**Ning Tian, Li Cheng \*, Yang Li** **, Xuan Li and Nan Xu**

School of Electrical and Information Engineering, Wuhan Institute of Technology, Wuhan 430205, China;
tianning0324@163.com (N.T.)
\* Correspondence: chengli8102@wit.edu.cn

**Abstract:** Color distortion, low contrast, and blurry details are the main features of underwater images, which can have adverse effects on their quality. To address these issues, a novel enhancement method based on color correction and multiscale fusion is proposed to improve underwater image quality, achieving color correction, contrast enhancement, and detail sharpening at different stages. The method consists of three main steps: color correction using a simple and effective histogram equalization-based method to correct color distortion, decomposition of the V channel of the color-corrected image into low- and high-frequency components using a guided filter, enhancement of the low-frequency component using a dual-interval histogram based on a benign separation threshold strategy, and a complementary pair of gamma functions; the fusion of the two versions of the low-frequency component to enhance image contrast; and finally, the design of an enhancement function to highlight image details. Comparative analysis with existing methods demonstrates that the proposed method achieves high-quality underwater images and favorable qualitative and quantitative evaluations. Compared to the method with the highest score, the average UIQM score of our method exceeds 6%, and the average UCIQE score exceeds 2%.

**Keywords:** color correction; contrast enhancement; underwater image enhancement

## 1. Introduction

The underwater image is a crucial means of understanding the underwater world, and it plays an essential role in underwater exploration, subaquatic (underwater) operations, and underwater target recognition. However, due to the selective absorption of light by water, underwater images lose their original color and exhibit a blue (green) hue, which reduces the clarity of the image [1]. Moreover, due to the complex scattering of the subaquatic medium, the image has a foggy effect that significantly reduces the clarity of the scene [2].

In order to tackle these challenges, scholars have devised a range of algorithms aimed at restoring and enhancing underwater images with the goal of improving their overall quality.

Image-restoration algorithms are built upon the foundations of underwater imaging models and can effectively handle degraded images. However, the effectiveness of these methods heavily depends on parameter estimation, and some methods require specialized underwater equipment, which increases costs. In recent years, some scholars have combined deep learning with underwater images, but this relies on synthesized underwater degraded image pairs and corresponding high-quality land images, often involving complex network architectures [3,4].

In contrast, underwater image-enhancement algorithms require less underwater-specific prior knowledge and aim to improve pixel quality by adjusting pixel intensities. Enhanced images exhibit higher contrast and richer detail, resulting in better visual effects. Common enhancement algorithms are predominantly based on spatial-domain- or fusion-based approaches.

Spatial-domain-based underwater image-enhancement algorithms effectively enhance image contrast. However, they may introduce red color casts and noise since color correction and noise handling are not explicitly considered. Fusion-based methods can improve image quality by reducing noise and enhancing details. However, these methods require the acquisition of multiple fusion images and the design of suitable fusion weights; otherwise, they may not yield positive results [3].

To address these issues, this paper introduces an innovative and reliable underwater image-enhancement algorithm. First, an approach for color correction utilizing histogram equalization is used to correct color distortions. Then, guided filtering is applied to decompose the V channel of the color-corrected image into low-frequency and high-frequency components. The low-frequency component is enhanced using a dual-interval histogram based on the benign separation threshold and a pair of complementary gamma functions. The two enhanced outputs of the low-frequency component are fused to enhance image contrast, and an enhancement function is designed to highlight image details. We summarize the innovation points of this article as follows:

- Color compensation is performed by combining the local and mean differences between the attenuation and non-attenuation channels of underwater degraded images. Based on this, a histogram correction technique based on histogram equalization is used to further correct the color of underwater images.
- Generate low-frequency components with different contrasts using dual-interval histograms and fuse the two versions of low-frequency components to enhance image contrast. Propose a function to highlight the high-frequency components of the V channel.

## 2. Related Works

In this section, we present a comprehensive review of relevant research, focusing on three main aspects: techniques for restoring underwater images, approaches for enhancing underwater images, and methodologies based on deep learning.

Underwater image-restoration technology estimates underwater imaging model parameters to obtain images before degradation. Underwater optical imaging-based [5–7] and polarization characteristics-based methods [8,9] consider the complexity of the underwater environment by constructing a specialized underwater imaging system, which yields results close to the ground truth. However, the limitations of these methods must also be considered—underwater optical imaging systems are sophisticated and costly as they require sophisticated hardware and capture equipment to obtain murky underwater images. He et al. [10] proposed the dark channel prior (DCP) for outdoor image dehazing. This is also applicable to underwater image processing since the degradation of images in foggy and underwater environments is caused by the irregular propagation of light in the medium, which share some similarities. Methods based on DCP have also been widely applied in underwater image processing. Galdran et al. [11] applied the idea of DCP to the red channel to improve the visibility of underwater images. Li et al. [12] utilized the histogram characteristics of images to process underwater images. Drews et al. [13] considered the degradation mode of underwater images and proposed a variant of DCP (UDCP). However, the method is not effective in the presence of white objects or artificial light. Peng et al. [14] used image ambiguity and underwater light attenuation to propose an underwater image-restoration method. Zhou et al. [15] used a color-line model to restore underwater images. However, since these recovery-based methods depend on subaquatic prior knowledge and specific model parameters, they may not be effective in certain water conditions. Liu [16] proposed a universal single-image-restoration algorithm by extending the dark channel prior (GDCP) to achieve restoration for different types of images; the method also demonstrated good performance in processing underwater images.

The enhancement algorithm enhances an image's quality by adjusting its pixels' intensity. Iqbal et al. [17] proposed an adaptive color-correction method for underwater images, where clustering algorithms are used to obtain the illuminant chromaticity of

the image and then a linear color transformation is applied to correct the color distortion. Ancuti et al. [18] fused two versions of images to enhance underwater images and videos. Fu et al. [19] proposed a Retinex-based method for correcting color and enhancing the contrast in underwater images using different approaches to enhance the two components of underwater images. They also proposed a two-step enhancement method [20] for single underwater images, using DCP to remove the color cast and then using histogram equalization and the Retinex algorithm to enhance the image details. Zhang et al. [21] utilized dual-interval histograms to improve the quality of underwater images based on color correction. Zhang et al. [22] proposed the minimum color loss theory, which combines underwater image attenuation maps to adjust the color and contrast of the image. Zhang et al. [23] designed a matrix for color correction and then used histogram technology to enhance the contrast of underwater images. These enhancement-based methods are generally more stable and effective at improving the contrast and clarity of underwater images.

Deep learning-based [24] underwater image-restoration and -enhancement methods have made significant progress in recent years, providing new possibilities for solving image quality problems in underwater environments. The method based on generative adversarial networks (GAN) performs well. By engaging generators and discriminators to learn from each other, these methods can generate clearer and more realistic underwater images, Yang et al. [25] improved the generation of adversarial networks by generating high-quality images through multiscale generators; Sun et al. [26] achieved underwater image enhancement in multiple scenarios by generating adversarial networks (UMGAN), which achieved unpaired image to image conversion between underwater turbid domains and underwater clear domains. Deep learning methods can improve image quality by learning the characteristics of underwater environments, thereby removing fog and dispersion from images. Li et al. [27] added underwater scenes before convolutional neural networks and constructed a synthetic underwater image dataset; Fu et al. [28] constructed a dual-branch network to address the issues of global distortion and contrast reduction in underwater images. The method based on deep learning relies on the synthesis of degraded underwater images and corresponding high-quality land images, so researchers focus on unsupervised learning and use the information from the images themselves for training, avoiding the need for a large number of datasets. Saleh et al. [29] achieved unsupervised underwater image enhancement by introducing adaptive uncertainty distributions into deep learning models.

## 3. Method of This Article

This article proposes a hybrid strategy to improve the quality of underwater images. This strategy consists of three main steps: removing color distortion; extracting the V-channel of the color-correction image and decomposing it into high-frequency and low-frequency components, enhancing the contrast of low-frequency components; and sharpening the details of high-frequency components (Figure 1).

### 3.1. Color Correction

Light travels differently underwater, resulting in a green (or blue) tint in underwater images due to the attenuation of blue and red light being more severe than green light. However, if the unique characteristics of color degradation in underwater scenes are not considered, this may cause additional color distortion, such as the introduction of red artifacts. To tackle this issue, our method uses an adaptive local compensation strategy to remove color distortion in underwater images, providing significant compensation for pixels with severe attenuation and limited compensation for others. The mathematical definition of this method is presented below.

$$I_{rc}(x,y) = I_r(x,y) + \left(I_g(x,y) - I_r(x,y)\right) \times \left(\bar{I}_g - \bar{I}_r\right) \tag{1}$$

$$I_{bc}(x,y) = I_b(x,y) + \left(I_g(x,y) - I_b(x,y)\right) \times \left(\overline{I}_g - \overline{I}_b\right) \tag{2}$$

where $I_r(x,y)$ represents the pixel value of the red channel at the pixel location $(x, y)$, $I_g(x,y)$ represents the pixel value of the green channel at the pixel location $(x, y)$, $I_b(x,y)$ represents the pixel value of the blue channel at the pixel location $(x, y)$, $I_{rc}(x,y)$ represents the pixel value of the red channel at the pixel location $(x, y)$ after compensation, $I_{bc}(x,y)$ represents the pixel value of the blue channel at the pixel location $(x, y)$ after compensation, $\overline{I}_g$ represents the average value of the entire green channel, and $\overline{I}_r$ represents the average value of all the red channel.

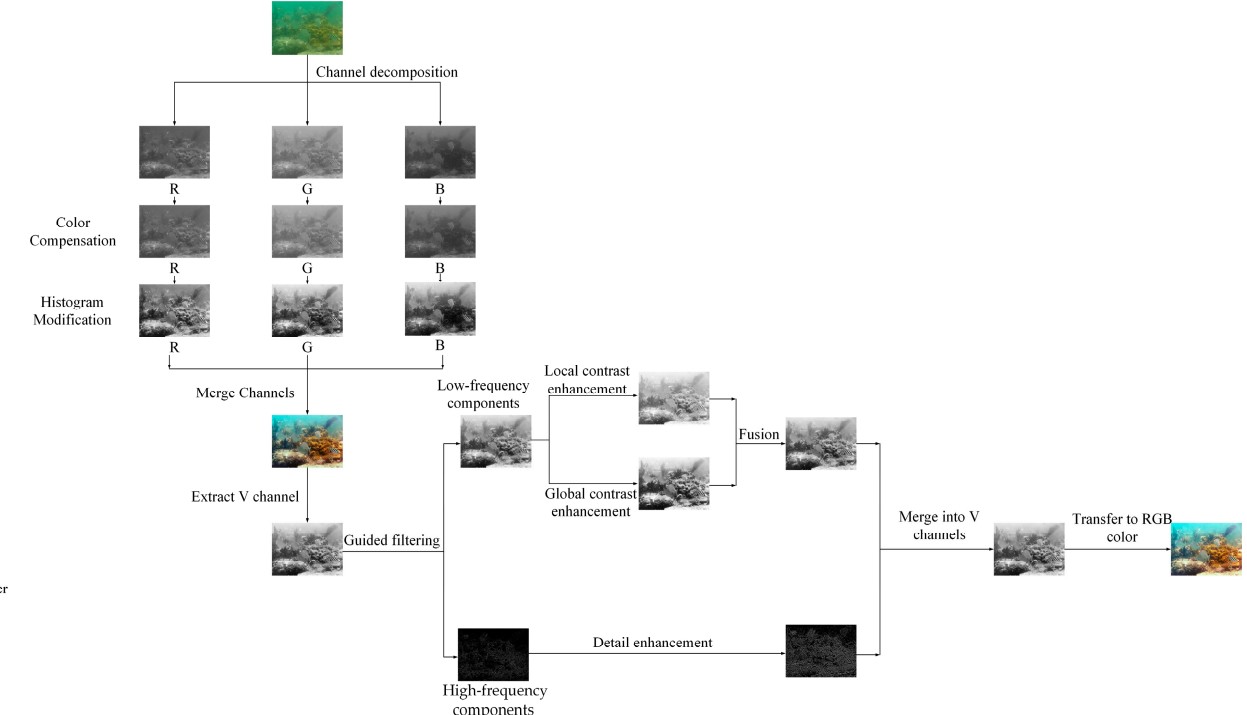

**Figure 1.** The framework of the proposed method. R, G and B are the color channels of RGB format images, respectively.

$(\overline{I}_g - \overline{I}_r)$ is crucial in describing the local attenuation of the red channel in comparison to the green channel, and pixels with significant attenuation of the red channel receive more compensation accordingly. The same principle applies to the blue channel.

After compensation processing, the grayscale value distribution of each color channel of the degraded underwater image remains uneven, and it is necessary to adjust the pixel value range of the image to enhance the visual effect and readability of the image. The linear stretching method can make the pixel value range more uniform, thereby making some detailed information that was originally compressed into a smaller pixel value range more visible. However, in some cases, linear stretching may also make the detailed information that was originally in the low pixel value range become blurrier or lost.

This paper proposes using the image histogram to subsequently correct the gray value distribution of each color channel in the underwater degraded image, even after the color compensation. The following section describes the process using the red channel as an example, with the same operation performed for the blue and green channels. The histogram of the red channel is defined as follows:

$$H(i) = n_i \ i \in [0, 255] \tag{3}$$

$n_i$ is the count of pixels in the compensated red channel with a pixel value of *i*.

$$H_C(i) = \begin{cases} th & if\ H(i) > th \\ H(i) & else \end{cases} \qquad (4)$$

In Equation (4), $H_C(i)$ represents the histogram after clipping, while *th* is the clipping threshold set in this paper, which can be mathematically expressed as follows:

$$th = mean(H(i)) + std(H(i)) \qquad (5)$$

In Equation (5), the term $mean(H(i))$ is the mean value of $H(i)$, while $std(H(i))$ is the standard deviation of $H(i)$. The next step is to assign the number of pixels in the clipped histogram. The mathematical definition is outlined below:

$$H_F(i) = H_C(i) + (\sum_{i=0}^{255} val(i))/255 \qquad (6)$$

$$val(i) = \begin{cases} H(i) - th & if\ H(i) > th \\ 0 & else \end{cases} \qquad (7)$$

After obtaining the redistributed histogram, its probability density function is expressed in Equation (8). For an image histogram, the probability density function (PDF) is obtained by dividing the frequency or probability of each pixel value by the length of the pixel value range, resulting in a probability density value corresponding to each pixel value.

$$PDF(i) = H_F(i)/sum \qquad (8)$$

The sum in this case refers to the total count of pixels in the red channel. Equation (9) is then used to calculate the cumulative distribution function (CDF) of the histogram. The CDF function is used to calculate the histogram equalization transformation function, so as to achieve the image gray-value equalization processing.

$$CDF(i) = \sum_{i=0}^{255} PDF(i) \qquad (9)$$

Finally, a color-corrected image is generated using Equation (10).

$$f = 255 \times CDF(i) \qquad (10)$$

### 3.2. Contrast Enhancement

Guided filtering [30] is an image filtering technique used to smooth images while preserving details and edge information. Compared to traditional filtering methods, the guided filtering has better noise reduction and image smoothing capabilities while preserving edges.

The principle of guided filtering is based on the following idea: it uses a guiding image, called the guidance image, to guide the filtering operation. The guidance image is typically an auxiliary image that is correlated with the input image, such as the grayscale image, gradient image, or other feature image derived from the original image. The guidance image provides guidance information about the image structure and features, allowing the filter to have a better understanding of the image content and edge information.

HSV (Hue, Saturation, Value) is a commonly used color space that is more consistent with human perception when describing colors. The HSV color space decomposes colors into three components: hue, saturation, and value. Hue represents the basic attributes of a color, which is the name of the color we perceive, such as red, green, blue, etc. Saturation represents the purity or depth of a color. Colors with higher saturation have a bright

appearance, while colors with lower saturation exhibit a darker or lighter effect. Value represents the brightness of a color.

The HSV color space is used in image processing, where the V channel represents the image's brightness. This paper enhances the contrast of the image in the V channel by decomposing it into low- and high-frequency components using guided filtering. The low frequency preserves the main changes in illumination, while the high frequency contains the image details.

$$V = V_s + V_t \tag{11}$$

In Equation (11), $V$ represents the brightness channel of the image, and $V_s$ and $V_t$ represent the low frequency and high frequency of the brightness channel, respectively.

Foreground and background regions in underwater images exhibit distinct characteristics and require different contrast-enhancement techniques. In our approach, we first use a benign separation threshold strategy to segment the underwater image into foreground and background sub-images. Subsequently, we perform contrast enhancement on these pairs of sub-images and utilize the Laplacian Pyramid Gaussian Pyramid to fuse the two versions of contrast-enhanced images. This effectively achieves a smooth transition between the enhanced images and eliminates discontinuities between them.

Through these steps, the visibility of underwater images is significantly improved while preserving the details and textures of the foreground and background areas.

### 3.2.1. Local Contrast Enhancement of Sub-Images

Most underwater images exhibit a bright foreground and dark background due to the characteristics of the underwater imaging system. Specifically, areas closer to the light source are brighter than areas farther away from the light source [31]. This paper uses Equation (12) to separate the threshold.

$$h = p_1(u_1 - u) - p_2(u_2 - u) \tag{12}$$

In Equation (12), $p_1$ and $p_2$ represent the probabilities of foreground and background pixels of the structural layer $V_s$, while $u_1$ and $u_2$ are the foreground and background pixels' average values, respectively. The $u$ represents the average value of pixels in $V_s$. The objective is to obtain the best threshold where the difference between the foreground and background is maximized. The larger the between-class variance $h$, the higher the difference between the two parts of the image.

We select T as the initial threshold to divide the image into two categories and then use Equation (12) as the fitness function to update h. When h is at its maximum, it means that T has been updated to the optimal threshold. Consequently, the input image can be divided into a background sub-image $V_s^D$ and a foreground sub-image $V_s^U$, as shown below:

$$V_s^D = \{V_s(x,y) | V_s(x,y) \leq T, \forall V_s(x,y) \in V_s\} \tag{13}$$

$$V_s^U = \{V_s(x,y) | V_s(x,y) > T, \forall V_s(x,y) \in V_s\} \tag{14}$$

where $V_s(x,y)$ represents the pixel value of the red channel at the pixel location $(x,y)$. The probability of sub-images $V_s^D$ and $V_s^U$ appearing in the sub-histogram of low and high intervals is defined as $P_D(V_s^D)$ and $P_D(V_s^U)$, respectively, and can be expressed as follows:

$$P_D(V_s^D) = H(V_s^D)/N_D \tag{15}$$

$$P_D(V_s^U) = H(V_s^U)/N_U \tag{16}$$

The frequency of grayscale occurrences in $H(V_s^D)$ and $H(V_s^U)$ are denoted as $V_s^D$ and $V_s^U$, respectively, where $N_D$ and $N_U$ are the total number of pixels in the background

and foreground sub-images, respectively. The CDFs of the background and foreground sub-images are defined as $C_D(V_s^D)$ and $C_D(V_s^U)$, respectively, and are expressed as follows:

$$C_D(V_s^D) = \sum_{V_s^D=0}^{T} P_D(V_s^D) \tag{17}$$

$$C_D(V_s^U) = \sum_{V_s^U=T+1}^{255} P_D(V_s^U) \tag{18}$$

Finally, to solve the low contrast of the image, the background sub-image and foreground sub-image are equalized using Equation (19), where $V_1$ refers to the structural layer $V_s$ after local contrast enhancement.

$$V_1 = \begin{cases} T \times C_D(V_s^D) & V_s^D \in [0, T] \\ (T+1) + [255 - (T+1)] \times C_U(V_s^U) & V_s^U \in [T+1, 255] \end{cases} \tag{19}$$

### 3.2.2. Global Contrast Enhancement of Sub-Images

One researcher [32] proposed a complementary relationship between the two functions in Equation (20) for enhancing low-light image contrast. The function curves are illustrated in Figure 2, indicating that function $y_2$ provides a more pronounced enhancement of pixels than function $y_1$.

$$\begin{cases} y_1 = 1 - (1-x)^{\gamma} & \gamma < 1 \\ y_2 = (1 - (1-x)^{1/\gamma})^{\gamma} & \gamma < 1 \end{cases} \tag{20}$$

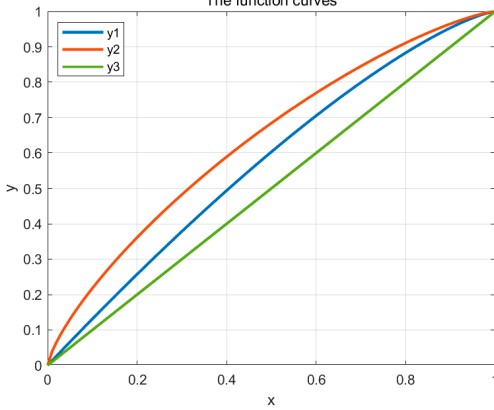

**Figure 2.** The function curves.

Therefore, we have designed a new global contrast-enhancement method for underwater images in Equation (21).

$$V_2(x,y) = \begin{cases} 1 - (1 - V_s(x,y))^{\gamma} & V_s(x,y) > T \\ (1 - (1 - V_s(x,y))^{1/\gamma})^{\gamma} & else \end{cases} \tag{21}$$

In Equation (21), $V_s(x,y)$ is the pixel value of $V_s$ at $(x,y)$, and $V_2(x,y)$ is the pixel value at $(x,y)$ in the image after contrast enhancement. This section uses the threshold obtained in local contrast enhancement to divide the input image for processing.

However, in this section, each pixel of the sub-image shares the same contrast-enhancement function, resulting in the global brightness improvement of the image. Figure 3 shows the image-enhancement effect of this section and section A. To visually display the impact, Figure 3a,b show the image converted from the space of the HSV to the RGB space.

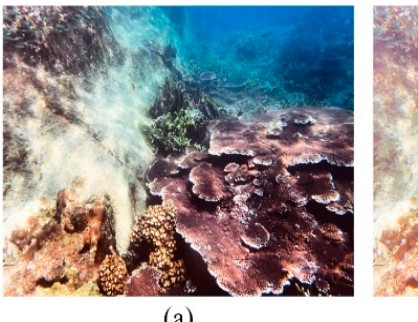 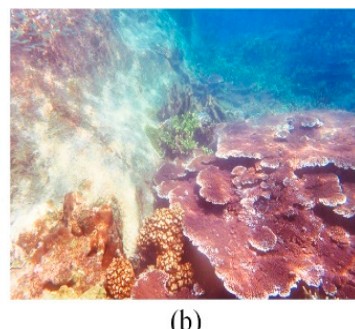

(a)             (b)

**Figure 3.** (**a**) is a local contrast-enhancement image and (**b**) is a global contrast-enhancement image.

### 3.3. Fusion

Multiscale fusion algorithms are commonly used in the fields of image processing and computer vision. They can fuse meaningful information from different scales together, enhance the features of input images, and thus improve the accuracy and efficiency of image processing and computer vision tasks. Multiscale fusion has shown excellent performance in applications such as remote-sensing image processing [33], defogging and rain removal [34], and high-resolution imaging. In this article, we generate a stable fusion framework that can increase image visibility and emphasize image details. Our framework is built on images with different contrasts generated from a single original image.

After the above processing, we obtained two enhanced image versions: $V_1$ and $V_2$. Because light exposure significantly affects image quality, we fused image regions with good brightness through exposure weight maps.

$$w_i(x,y) = \exp\left\{-\frac{(V_i(x,y)-0.5)^2}{2 \times 0.25^2}\right\} \tag{22}$$

The formula for calculating the normalized weight map for fusion is as follows:

$$\overline{w}_i(x,y) = \frac{w_i(x,y)}{\sum_i w_i(x,y)} \tag{23}$$

where $V_i(x,y)$ represents the $i$th input image at $(x,y)$. We use the pyramid fusion method. The resulting enhanced image is obtained as:

$$V_{\text{es}}(x,y) = \sum_l \left(\sum_{i=1}^{2} G^l(\overline{w}_i(x,y))L^l(V_i(x,y))\right) \tag{24}$$

where $l$ represents the number of layers of the pyramid, $G(\cdot)$ represents Gaussian operators, and $L(\cdot)$ represents Laplacian operators, respectively.

### 3.4. Detail Enhancement

The proposed method in this paper presents a simple approach for enhancing the details in underwater images. The texture details are adaptively amplified, and the blurred details are significantly improved using an enhancement coefficient K. Through extensive experiments and considering the influence of parameters on visibility and the human visual system, $\lambda$ and $\sigma$ in Equation (25) are set to be 8 and 1.

$$K = 1 + \lambda \exp\left(\frac{-|V_t(x,y)|}{\delta}\right) \tag{25}$$

$V_t(x,y)$ represents the pixel value of $V_t$ at the pixel location $(x, y)$, while $V_{et}(x,y)$ represents the pixel value of the texture layer at the pixel location $(x, y)$ after enhancement.

$\varepsilon$ is the parameter used to avoid the dense noise of enlarging the underwater image. In our experiment, $\varepsilon$ is 0.005. Equation (26) is used to enhance the details.

$$V_{et}(x,y) = \begin{cases} 0 & if\,|V_t(x,y)| < \varepsilon \\ \text{K} \times V_t(x,y) & else \end{cases} \tag{26}$$

Finally, the enhanced input of the *V* channel is as follows:

$$V_{final} = V_{es} + V_{et} \tag{27}$$

The final step involves converting the enhanced *V* channel image and the H, S channel image from the HSV color space to the RGB color space to obtain the improved underwater image.

## 4. Results

In this section, we perform qualitative and quantitative evaluations to assess the effectiveness of the proposed algorithm. We compare our algorithm with several existing advanced underwater image-enhancement methods, namely ARC [11], Fusion [18], GDCP [16], IBLA [14], TS [20], UDCP [13], MLLE [22], and ACCDO [23]. We used a UIEB [35,36] dataset with 950 real-world underwater images for evaluation, of which 890 images had corresponding reference images, referred to as UIEBR, and the remaining 60 underwater images were referred to as UIEBC. Table 1 shows the methods we compared.

**Table 1.** These methods are replaced with abbreviations later in this article.

| Method | Abbreviation |
|---|---|
| Automatic red-channel underwater image restoration [11] | ARC |
| Underwater depth estimation and image restoration based on single images [13] | UDCP |
| Underwater image restoration based on image blurriness and light absorption [14] | IBLA |
| Generalization of the dark channel prior for single image restoration [16] | GDCP |
| Color balance and fusion for underwater image enhancement [18] | Fusion |
| Two-step approach for single underwater image enhancement [20] | TS |
| Underwater Image Enhancement via Minimal Color Loss and Locally Adaptive Contrast Enhancement [22] | MLLE |
| Underwater Image Enhancement by Attenuated Color Channel Correction and Detail Preserved Contrast Enhancement [23] | ACCDO |

### 4.1. Qualitative Evaluation

For the qualitative evaluation, we initially chose two types of underwater images from UIEBR: blue and green. Due to space limitations, we presented only a selection of images. The enhanced images produced by various algorithms are shown in Figures 4 and 5.

For the blue image presented in Figure 4, the GDCP, IBLA, and UDCP algorithms significantly improved the saturation, visual effect, and visibility, but their color-correction performance was unsatisfactory. The local features of the image processed by TS are prominent. However, it does not effectively address the color deviation, such as in the processing images of B2. The processing results of this fusion method are closest to the reference image. ARC employed effective color-correction methods, but the enhancement of darker areas of the underwater image was not prominent.

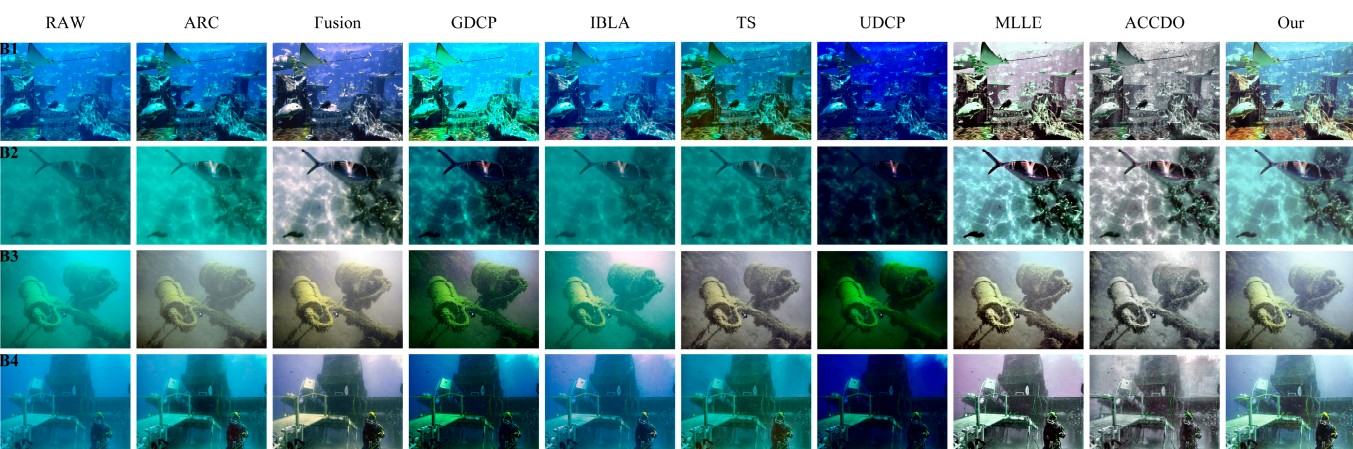

**Figure 4.** Enhancement effects of different methods, including ARC, Fusion, GDCP, IBLA, TS, UDCP, MLLE, ACCDO, and our method. Enhancement effects of different methods on blue images, including ARC, Fusion, GDCP, IBLA, TS, UDCP, MLLE, ACCDO, and our method.

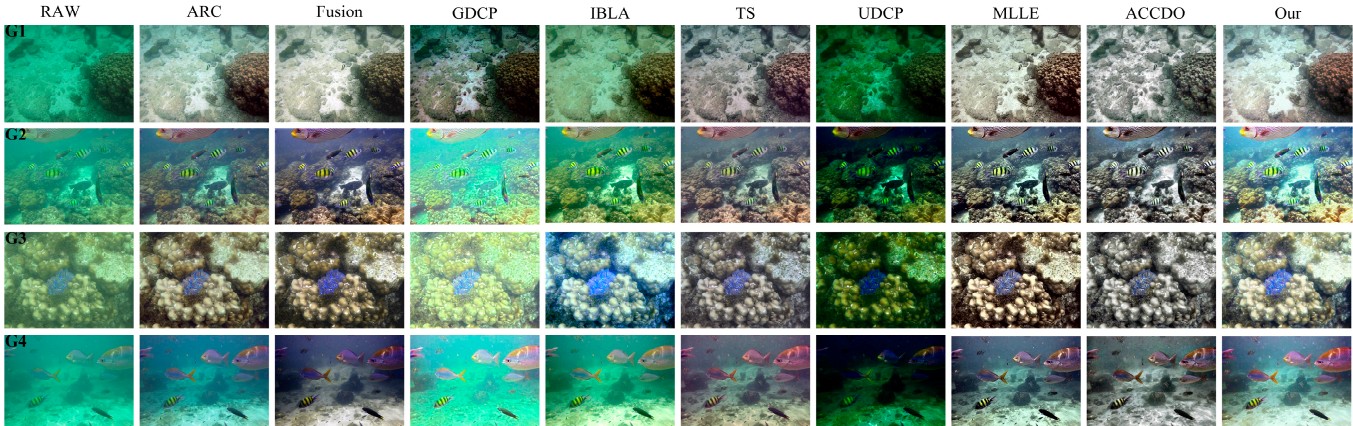

**Figure 5.** Enhancement effects of different methods, including ARC, Fusion, GDCP, IBLA, TS, UDCP, MLLE, ACCDO, and our method. Enhancement effects of different methods on green images, including ARC, Fusion, GDCP, IBLA, TS, UDCP, MLLE, ACCDO, and our method.

For the green image shown in Figure 5, the effect of UDCP cannot meet the requirements well. IBLA has achieved excellent visibility and color correction. However, the treatment of G1 and G2 is unsatisfactory. Among the compared algorithms, ARC, Fusion, and TS exhibit better performance concerning improving visibility and correcting color for the green images. However, their contrast-enhancement effect is somewhat limited, especially in the case of G2 images. The GDCP significantly enhances underwater image contrast, although the color correction is nonideal. The enhanced images obtained by our method have good results in terms of color, contrast, and details.

### 4.2. Quantitative Evaluation

In the quantitative evaluation, we selected three full-reference indicators to assess the quality of the underwater image shown in the figure: edge intensity (AG) [3], information entropy (IE) [3], and patch-based contrast quality index (PCQI) [35]. Furthermore, we employed two non-reference indicators, namely the underwater image quality measurement (UIQM) [37] and the underwater color image quality-evaluation index (UCIQE) [38], to evaluate the quality of the underwater images.

AG is mainly used to represent the clarity of images, while IE is used to describe the average information content of underwater images. PCQI is used to evaluate the local contrast of underwater images. The UIQM is a comprehensive evaluation indicator

that includes color, clarity, and contrast. The UCIQE is a linear combination of color concentration, saturation, and contrast. We strive to comprehensively evaluate the existing methods and the methods presented in this article through these indicators.

To further demonstrate the effectiveness of our algorithm, we evaluated the entire UIEB dataset, including UIEBR and UIEBC. Tables 2 and 3 present the performance indicators of this method and other methods. Our algorithm achieves the highest scores in IE, the UIQM, and the UCIQE on UIEBR and UIEBC, while its AG and PCQI scores are lower than those of the MLLE algorithm. Nonetheless, our algorithm still demonstrates competitiveness, and compared to mainstream methods, the MLLE algorithm performs better in handling the details and local contrast of degraded images, which is an area for improvement in our future work.

**Table 2.** Comparison of indicators for different methods in UIEBR. The red value represents the best, while the blue value represents the second best.

| UIEBR | ARC | Fusion | GDCP | IBLA | TS | UDCP | MLLE | ACCDO | OUR |
|---|---|---|---|---|---|---|---|---|---|
| AG | 4.835 | 6.319 | 7.221 | 5.989 | 7.194 | 5.207 | **12.913** | 9.522 | **12.663** |
| IE | 7.187 | 7.413 | 7.316 | 7.267 | 7.253 | 6.557 | 7.580 | **7.662** | **7.727** |
| PCQI | 1.013 | 1.066 | 1.045 | 1.074 | 1.148 | 0.814 | **1.221** | **1.191** | 1.044 |
| UIQM | 3.214 | 3.516 | 2.568 | 2.560 | 3.245 | 2.405 | 2.607 | **3.520** | **3.745** |
| UCIQE | 0.563 | 0.588 | **0.610** | 0.604 | 0.601 | 0.584 | 0.605 | 0.555 | **0.620** |

**Table 3.** Comparison of indicators for different methods in UIEBC. The red value represents the best, while the blue value represents the second best.

| UIEBC | ARC | Fusion | GDCP | IBLA | TS | UDCP | MLLE | ACCDO | OUR |
|---|---|---|---|---|---|---|---|---|---|
| AG | 3.139 | 4.512 | 4.848 | 4.302 | 5.070 | 3.055 | **7.734** | 6.512 | **6.347** |
| IE | 7.057 | 7.253 | 7.122 | 6.998 | 7.215 | 5.638 | 7.312 | **7.519** | **7.624** |
| PCQI | 0.992 | 0.998 | 0.954 | 1.011 | 1.052 | 0.801 | **1.086** | **1.071** | 0.914 |
| UIQM | 2.149 | **2.175** | 1.882 | 1.841 | 2.386 | 1.621 | 1.648 | 1.952 | **2.214** |
| UCIQE | 0.536 | 0.572 | 0.565 | **0.591** | 0.574 | 0.520 | 0.579 | 0.549 | **0.596** |

### 4.3. Running Time

The runtime is an important criterion for determining whether an algorithm can process in real time. We have chosen underwater images with different resolutions to detect the runtime of different algorithms. All the algorithms mentioned in the article are executed on the same PC and MATLAB (2018b). The computer is configured with Intel i7 118000H, 2.3 GHz, and 16 GB of running memory. The running time of different algorithms is shown in Table 4. With the improvement of image resolution, UDCP and IBLA require a longer processing time. The TS, GDCP, and MLLE algorithms have excellent performance in terms of runtime, and the algorithm proposed in this article has good competitiveness.

**Table 4.** Comparison of runtime for different methods. The red value represents the best, while the blue value represents the second best.

| Resolution | ARC | Fusion | GDCP | IBLA | TS | UDCP | MLLE | ACCDO | OUR |
|---|---|---|---|---|---|---|---|---|---|
| 500 × 375 | 1.027 | 1.499 | 0.276 | 11.169 | 0.128 | 6.664 | 0.159 | 0.488 | 0.121 |
| 640 × 480 | 1.678 | 2.476 | 0.285 | 21.696 | 0.178 | 12.630 | 0.226 | 0.811 | 0.175 |
| 850 × 564 | 2.645 | 3.833 | 0.373 | 35.711 | 0.254 | 18.946 | 0.362 | 1.251 | 0.297 |
| 1280 × 720 | 5.060 | 7.576 | 0.551 | 62.228 | 0.474 | 40.256 | 0.823 | 2.195 | 0.558 |
| Ave | 2.603 | 3.846 | 0.396 | 32.701 | **0.259** | 19.624 | 0.393 | 1.186 | **0.288** |

### 4.4. Detail Analysis

The information hidden in the details of an image is often critical, and bright image details are vital for practical applications such as underwater target detection [39] and

tracking. Therefore, an excellent underwater image-enhancement algorithm should correct the color, improve contrast, and enhance image details. Figure 6 shows the enhanced images and the details produced by different algorithms. The MLLE and ACCDO algorithms have achieved satisfactory results. ARC, Fusion, and TS algorithms enhance image details to a certain extent. The pattern of the small fish in Figure 6 is clearer than in the original image, but the contrast is somewhat lacking; GDCP and IBLA have improved contrast, but there is no enhancement of details. The algorithm in this article takes into account both details and contrast.

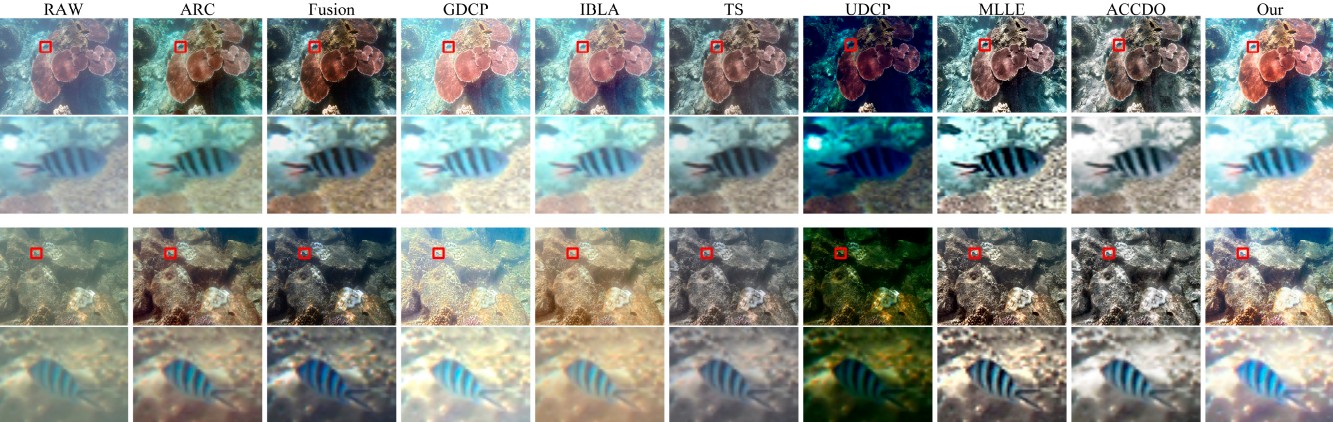

**Figure 6.** Enhancement results of different methods and the red box represents the corresponding locally enlarged image. From left to right are raw underwater images, and the result of ARC, Fusion, GDCP, IBLA, TS, UDCP, MLLE, ACCDO, and the proposed method.

*4.5. Application Test*

We conducted experiments on two visual tasks—low-light image enhancement and local feature-point matching—to demonstrate the positive role of our method in vision.

The visibility and contrast of the image are often reduced in low-light environments, making it difficult to extract valuable information or make further use of the image [40]. Here, we apply our method to images captured in low-light environments, as shown in Figure 7. Our approach significantly improves visibility and contrast in low-light images.

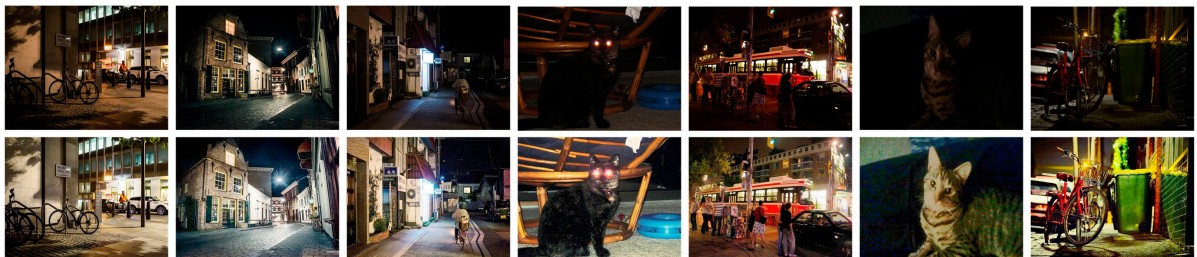

**Figure 7.** The upper column is the original image, and the lower column is the image processed by our method.

To demonstrate that the images processed in this article can be applied to visual matching tasks, we used the local feature-point matching algorithm (SIFT) to establish the corresponding relationship between two similar scenes. We apply this algorithm to a pair of real underwater images and their improved matching results. Figure 8 shows the local feature-point matching results. In Figure 8a,b, the correctly matched feature points in the original image pair are 4 and 87, respectively. Figure 8c,d show the number of feature points matched in the image pairs processed by our proposed algorithm, which are 26 and 140, respectively. This indicates that our method has played a positive role in image preprocessing.

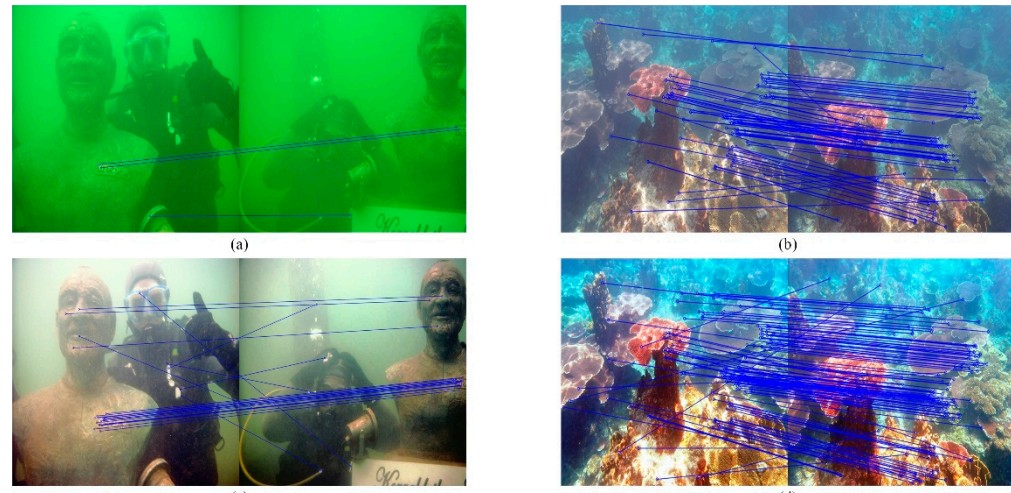

**Figure 8.** Local feature points match by using the SIFT [41]. (**a**,**b**) are the feature-point matching results of the original image, (**c**,**d**) are the feature-point matching results of the enhanced image using this method.

## 5. Conclusions

We propose a method for enhancing underwater images using color correction and multiscale fusion. The test results on mainstream underwater image datasets show that the enhanced images obtained by this method have high quality.

Although our method has some superior performance, it also has some limitations. On the one hand, our method has a less significant enhancement effect on delicate image textures, which may be due to the neglect of detail processing in the fusion framework. On the other hand, simple threshold filtering has been chosen for underwater noise, which may fail when facing complex underwater scenes and also eliminates small image details. Therefore, we will consider researching and addressing these issues in our future work, attempting to find a more excellent fusion strategy that takes into account the improvement of contrast and detail. The denoising of underwater images is also a direction worth focusing on.

**Author Contributions:** Conceptualization, N.T. and L.C.; methodology, N.T. and L.C.; software, N.T.; validation, L.C. and Y.L.; formal analysis, N.X.; investigation, N.T.; resources, L.C.; writing—original draft preparation, N.T.; writing—review and editing, L.C.; project administration, L.C.; funding acquisition, X.L. and L.C. All authors have read and agreed to the published version of the manuscript.

**Funding:** This research was funded by the Natural Science Foundation of Hubei Province of China under Grant (No. 2022CFB776), and the Foundation of Wuhan Institute of Technology under Grant (No. CX2022160).

**Institutional Review Board Statement:** Not applicable.

**Informed Consent Statement:** Not applicable.

**Data Availability Statement:** These data can be found here: https://li-chongyi.github.io/proj_benchmark.html (accessed on 1 June 2023).

**Conflicts of Interest:** The authors declare no conflict of interest.

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
