# Peer review of "Enhancing Underwater Images via Color Correction and Multiscale Fusion"

_applsci, doi:10.3390/app131810176_

Round 1

Reviewer 1 Report

Interesting topic and method for underwater enhancement. I have the following major issues, please carefully address them before publication.

1.      I suggest the author add a few sentences to review the progress of deep-learning based image restoration/enhancement method. Thus, the readers can have a comprehensive insight of the tropic. A quick google search shows me the following related works: [1] Jiang et al. Degrade is upgrade: Learning degradation for low-light image enhancement; [2] A Saleh et al. Adaptive uncertainty distribution in deep learning for unsupervised underwater image enhancement

2.      The idea of multi-scale fusion has already been used in various low-level vision tasks [1] [2], please give us more evidence from published works to demonstrate the importance of multi-scale features in improving the quality of images. [1] Xiao et al. Satellite Video Super-Resolution via Multiscale Deformable Convolution Alignment and Temporal Grouping Projection; [2] Multi-scale progressive fusion network for single image deraining

3.      The resolution of figure 2 is too low. Please provide a high-resolution version.

4.      Can you provide some results on real underwater scenes?

5.      Missing disadvantages and future guidelines in the conclusion section.

Please refer to the detailed comments.

Author Response

Thank you very much for your feedback. We have made the modifications according to your requirements.
1. We have added a summary of deep learning in the relevant work section of the article, which can be viewed in bold blue.
2. We have added a brief overview of multi-scale fusion in the image fusion section, explaining the important role of multi-scale fusion in image processing. You can see the bold blue section.
3. We have replaced Figure 2 with a higher resolution image.
4. The UIEB dataset is a real underwater image, as it is difficult to capture underwater images, we do not have our own actual captured images.
5. In the conclusion section of the article, we have shown the shortcomings of our work, analyzed the possible reasons, and discussed our future work. This can be seen in the bold blue section.

Reviewer 2 Report

In this manuscript, the authors focus on enhancing underwater images via color correction and Multiscale Fusion. Where the authors mention the color correction using a simple and effective histogram equalization-based method to correct color distortion, decomposition of the V channel of the color-corrected image into low and high-frequency components using a guided filter, enhancement of the low-frequency component using a dual-interval histogram based on a benign separation threshold strategy and a complementary pair of gamma functions, fusion of the two versions of the low-frequency component to enhance image contrast. In fact, fusion technology in enhancing underwater images is very popular. Here, I provide a paper, the idea of this manuscript is very similar to the following paper [1].

[1] Zhou, Jingchun, Dehuan Zhang, and Weishi Zhang. "Multiscale fusion method for the enhancement of low-light underwater images." Mathematical Problems in Engineering (2020): 1-15.

1) The authors should provide a statement to show the difference between this manuscript and the paper [1].

2)In the contribution part, the authors write down four contributions. Where the first and second cannot be as a contribution. Please summarize four contributions into two contributions. Moreover, please state the novel technology authors proposed, don’t use vague language to describe.

3) In Figure 1, Please state the technique details at each step in Figure 1 and the caption of Figure 1. Do not only show the resulting image.

4)Please introduce the advantages and limitations of your proposed methods compared with other existing methods.

Minor editing of the English language required

Author Response

Thank you for your feedback. 

1.We have carefully reviewed Zhou's article and compared it with our work. The ideas and framework of the two articles are similar, but the handling of each step is not the same.
In terms of color correction, Zhou et al. combined YCbCr with RGB color space. Our work involves color compensation in the RGB space and then stretching using histogram technology. The two articles are completely different here.
In terms of improving contrast:
(1)We transferred the image into the HSV color space, decomposed the V channel into high-frequency and low-frequency, and fused the V channel. Zhou et al. conducted the fusion in the RGB space.
Our work is to fuse the high-frequency components of local and global contrast enhancement. Zhou et al. fused the low-frequency and high-frequency information after gamma correction with the high-frequency information after dual interval equalization of the gradient field.
(2) The weights of fusion are not the same. We are a single channel fusion that relies on exposure weights.
(3) For the improvement of details, we have designed our own enhancement function.

2.We have rewritten the contribution section of the article and highlighted it in green font.

3.We have redrawn Figure 1, please take a look.

4.Compared with existing enhancement methods, the method proposed in this article proposes a simple and stable color correction technique, which can avoid reflecting red artifacts on heavily degraded underwater images, but does not effectively enhance small details,
For example, small protrusions on stones in the water will be smoothed, and the fusion of single channels takes into account contrast, but the handling of details is not outstanding. Compared to existing restoration methods,
The method in this article does not involve complex prior knowledge of underwater environments. Starting from the image itself, the pixel intensity of the image is modified to enhance the image, which is suitable for processing large-scale and different underwater scenes,
However, this article only considers the propagation mode of light in water and does not introduce underwater imaging models like existing restoration methods, although the enhancement results have shown good performance in indicators and other application tests,
But the difference between the enhanced results and the real land image is not known.

Reviewer 3 Report

  The authors present a new image processing method to enhance blurry underwater images. The proposed method improves the quality of underwater photos.  The paper clearly states the existing literature with image processing methodologies applied to the same problem and is presented concisely.   1. Methodology The proposed algorithm is interesting even though the underlying mathematical constructs are from existing literature based on light properties underwater and how they affect different colours. The authors have presented an exciting algorithm designed from existing methods' deficiencies. I would not say it is novel, but certainly a suitable method. The algorithm overall includes (a) Colour correction (b) Contrast enhancement (c) Fusion of features, and (d) Detail enhancement.    Even though the core is around Colour correction and Contrast enhancement, the Fusion and Detail enhancement steps are more cosmetic to correct for the initial contrast enhancement done. The latter two steps mostly use existing methodologies.    2. Results Compared to other techniques, the method shows good overall results with various standard tests. In terms of enhancing image quality mentioned in the introduction, it would also be good to have a measure of SNR, which is one primary quality measure.   3. Detail Analysis (section 4.3) This section needs more qualitative content. The examples are shown, but I do not see a proper analysis done. It is worth mentioning that apart from pure image processing techniques, there are machine learning, specifically deep learning techniques, applied to the same problem. The analysis could cover the result comparison with at least one deep learning method. The authors have not also commented on the computational complexity of the method, which could be compared with both deep learning and computer vision-based methods already mentioned in the analysis.    The computational complexity is fascinating because it will determine if this can even be applied to real-time video processing. Such methods are generally applied to offline analysis, but there is also enormous potential for online/real-time analysis. 

4. Application Test The authors have shown reasonable examples with application tests. The use of crucial point detection is a worthy example to showcase.  The authors have not mentioned publishing their code which is encouraged.   

Author Response

Thank you very much for your feedback, which has greatly benefited us. We have made modifications based on your feedback, but some work has not been carried out smoothly.
The code for the paper is being integrated, and we will provide our source code afterwards.
2. Running time is an important criterion for determining whether an image can be processed in real-time. We tested underwater images of different resolutions, and the method proposed in this article showed great competitiveness. I think this can be applied in real-time.
3. Yes, we wanted to evaluate SNR, but there were unfounded results in the evaluation. The SNR of images with better indicators and visual effects received lower ratings. After reviewing relevant papers, we found that no appropriate suggestions were given for SNR evaluation, which may also be due to our incorrect evaluation method. We would like PSNR to replace it again, but unfortunately there is no matching terrestrial reference image.
4. You mentioned that the method used in this article lacks quantitative evaluation in detail analysis. We have reviewed the literature and found that the data indicators for detail analysis may require researchers to have a certain professional background. At the same time, there is a lack of quantitative indicators for underwater image detail analysis, which may be a research direction in the future. We try to demonstrate subjective effects as much as possible, although this is not rigorous enough.
5. Deep learning is currently a mainstream processing method. We contacted relevant authors to try to obtain source code and datasets for training, but unfortunately did not receive a response. At the same time, deep learning also requires some professional knowledge and good hardware foundation, which are currently lacking. Thank you very much for your feedback. In the future, we will conduct research on deep learning in underwater image processing, Shift the focus towards deep learning.

Round 2

Reviewer 1 Report

The authors have addressed all my concerns. With the current version, it is good enough for publication.

None

Author Response

This is the modified version.

Reviewer 2 Report

The author answers my questions.

The author answers my questions.

Author Response

This is the modified version
